# Two-dimensional Thouless pumping of light in photonic moiré lattices

Peng Wang [1], Qidong Fu [1], Ruihan Peng [1], Yaroslav V. Kartashov [2,3], Lluis Torner [2,4], Vladimir V. Konotop [5,6] & Fangwei Ye [1] ✉

Continuous and quantized transports are profoundly different. The latter is determined by the global rather than local properties of a system, it exhibits unique topological features, and its ubiquitous nature causes its occurrence in many areas of science. Here we report the first observation of fully-two-dimensional Thouless pumping of light by bulk modes in a purpose-designed tilted moiré lattices imprinted in a photorefractive crystal. Pumping in such unconfined system occurs due to the longitudinal adiabatic and periodic modulation of the refractive index. The topological nature of this phenomenon manifests itself in the magnitude and direction of shift of the beam center-of-mass averaged over one pumping cycle. Our experimental results are supported by systematic numerical simulations in the frames of the continuous Schrödinger equation governing propagation of probe light beam in optically-induced photorefractive moiré lattice. Our system affords a powerful platform for the exploration of topological pumping in tunable commensurate and incommensurate geometries.

Transfer of matter, charge, or spin, carried by quantum or classical extended wave packets depends on the global characteristics of the system in which it occurs. Topological charge pumping that transfers current in a quantized manner was discovered by Thouless[1] in condensed-matter physics. It has been recognized as a phenomenon of paramount significance, because of its fundamental relation to general underlying topological properties of the system, which are also at the origin of the quantum Hall effect[2]. More broadly, as a general wave phenomenon, topological pumping is appreciated as important for many branches of physics. To date, it has been observed in semiconductor quantum dots[3], in spin systems[4], in one-[5,6] and two-dimensional[7] optical waveguide arrays, in cold bosonic[8–11] and fermionic[12] atoms in moving optical lattices, in acoustic metamaterials[13], and in dissipative arrays of plasmonic waveguides[14]. Most of the above realizations were conducted in one-dimensional (1D) settings. Only a few involved 2D settings where the topological

transport is carried out by edge states[7]; hence, there is a prescribed direction dictated by the boundaries. Nevertheless, in a fully 2D geometry, in addition to the quantized nature of the amount of charge or matter transferred, the direction of transport is also characterized by a non-vanishing quantized angle with respect to the direction of pumping, and, importantly, the very transport is carried out by bulk modes, rather than by edge or boundary modes. Even though the first theoretical proposals for 2D topological pumping in bilayer graphene systems with mutually displaced layers are available[15–17], the only experimental observation of fully 2D topological pumping so far was achieved in a system of tightly confined cold atoms[10], where the direction of collective atomic motion is characterized by a small angle with respect to pumping direction, determined by a combination of the first and second Chern numbers.

The experimental realization of Thouless pumping carried out by bulk modes is a challenging task because in most systems linear wave

[1]State Key Laboratory of Advanced Optical Communication Systems and Networks, School of Physics and Astronomy, Shanghai Jiao Tong University, Shanghai 200240, China. [2]ICFO-Institut de Ciencies Fotoniques, The Barcelona Institute of Science and Technology, 08860 Castelldefels, Barcelona, Spain. [3]Institute of Spectroscopy, Russian Academy of Sciences, Troitsk, Moscow 108840, Russia. [4]Universitat Politecnica de Catalunya, 08034 Barcelona, Spain. [5]Departamento de Física, Faculdade de Ciências, Universidade de Lisboa, Campo Grande, Ed. C8, Lisboa 1749-016, Portugal. [6]Centro de Física Teórica e Computacional, Faculdade de Ciências, Universidade de Lisboa, Campo Grande, Ed. C8, Lisboa 1749-016, Portugal. ✉e-mail: fangweiye@sjtu.edu.cn

packets undergo strong spreading due to dispersion (diffraction in photonic systems). Counteracting such limitation requires, on the one hand, controlling the gapped spectrum of a periodically varying lattice potential at any instance of the wavepacket evolution, and, on the other hand, simultaneously inhibiting the wavepacket spreading to an extent that allows observations to be conducted during a few cycles of the adiabatic evolution. In atomic systems such control is possible in the Mott insulator phase in a sufficiently deep lattice and in the presence of inter-atomic interactions[10], or by confining an atomic cloud in a suitable external trap[18]. However, 2D pumping of pure bulk modes of light (or other waves) in linear unconfined systems, where the diffraction (or dispersion) is stronger than in earlier 1D experiments or in 2D settings with boundary modes, was never addressed.

By and large, the shape of the refractive index landscape of an optical medium determines the set of control parameters available for tuning the propagation properties of wave packets in the system. In conventional periodic media, these are the symmetry properties defining the bandgap structure of the wave spectra. Fundamentally new possibilities and, therefore, evolution regimes appear when a lattice is formed by the superposition of two (or more) periodic sublattices that are mutually rotated, twisted or shifted with respect to each other so that a moiré pattern forms. Moiré heterostructures greatly enrich the transport and localization properties of excitations propagating in them and lead to the appearance of new types of phenomena that are under intense current investigation in 2D materials[19,20]. A central physical mechanism affecting the electron propagation in the moiré heterostructures is the appearance of flat bands[21,22]. Similarly, the spectrum of a photonic mono-layered moiré lattices contains a considerable number of nearly flat bands. This makes possible spatial localization of linear light beams manifesting nearly diffraction-free propagation[23,24].

Here we predict theoretically, and report numerically and experimentally, genuinely 2D topological pumping of a light beam without using any kind of the transverse confinement and without employing nonlinearity as a self-confining mechanism. The key physical mechanism that enables the simultaneous control of the gapped spectrum and inhibition of diffraction of the modes is the enhanced flatness of the excited bands of a new, specially prepared photonic moiré lattice that is optically imprinted in a photorefractive crystal. Using 3D rotations of the two constitutive sublattices (beyond usual twists that generate standard moiré patterns) we create a moiré pattern adiabatically sliding in the course of propagation along the crystal. In this way we induce fully 3D moiré lattice that periodically changes in all three dimensions (see schematics in Fig. 1a). Such a re-configurable pattern allows the observation of topological 2D Thouless pumping of a weakly diffracting signal beam accomplished by fully 2D bulk modes. To the best of our knowledge, the photonic system based on two tilted and mutually twisted sublattices forming dynamical moiré lattice that nevertheless periodically restores its profile in the course of evolution, has never been addressed. Importantly, such 2D dynamically changing lattices may feature band structures that do not exhibit band crossings at any instants of evolution. As shown below, the creation of such moiré lattices enables 2D Thouless pumping in a new regime, namely, directional pumping in a fully 2D setting without relying on any confining mechanisms.

## Results

### Sliding moiré lattices

For the realization of the mutually tilted moiré lattices we employ the technique of optical lattice induction in photorefractive crystals[25,26]. The optical lattice induced by the ordinary-polarized light beams, creates a refractive index modulation for the extraordinary-polarized signal beam whose evolution in the paraxial approximation is governed by the Schrödinger equation for the dimensionless field amplitude $\psi(\boldsymbol{r}, z)$:

$$i\frac{\partial\psi}{\partial z} = H\psi, \qquad \text{with} \qquad H = -\frac{1}{2}\nabla^2 - U(\boldsymbol{r},z). \qquad (1)$$

Here $\nabla = (\partial/\partial x, \partial/\partial y)$, the transverse radius-vector $\boldsymbol{r} = (x, y)$ is scaled to the actual beam width $r_0 = 9\,\mu\text{m}$ used in the experiments; the propagation distance $z$ (which in our system plays the role that time does in quantum systems) is scaled to the diffraction length $2\pi n_e \lambda$, $n_e \approx 2.2817$ is the refractive index of the homogeneous SBN:61 crystal for extraordinarily polarized light. The induced optical potential experienced by the signal beam is given by $U(\boldsymbol{r}, z) = -E_0/(1 + I(\boldsymbol{r}, z))$, where $E_0 > 0$ is the applied dimensionless dc field, and $I(\boldsymbol{r}, z)$ is the intensity distribution of laser beams imprinting the lattice, which in our case depends on the transverse, $\boldsymbol{r}$, and longitudinal, $z$, coordinates. The periodic dependence on $z$ at small tilt angle $\alpha \ll 1$ emulates the adiabatic pumping.

The Pythagorean moiré lattice shown in Fig. 1 is a superposition of two square sublattices $V(R(\theta)\boldsymbol{r})$ and $V(\boldsymbol{r} - \alpha z\mathbf{j})$, where $R(\theta)$ is the standard 2D rotation operator by a twist angle $\theta$. In our experiment each of the sublattices were induced by four interfering plane waves (see Methods). The second sublattice has a relative amplitude $p$ with respect to the first one. When the twist angle is given by $\theta = \arctan(2mn/(m^2 - n^2))$ with integer $m$ and $n$, in the absence of a tilt, i.e., when $\alpha = 0$, it corresponds to the Pythagorean angle associated with the Pythagorean triple $(m^2 - n^2, 2mn, m^2 + n^2)$. The resulting moiré lattice $I(\boldsymbol{r}, z) = |V(R(\theta)\boldsymbol{r}) + e^{ik_z(\alpha y - \alpha^2 z/2)} p V(\boldsymbol{r} - \alpha z\mathbf{j})|^2$, where $k_z$ is the $z$-component of the wavevector (see Supplementary materials), is commensurate for $\alpha = 0$, i.e., it is exactly periodic in the $(x, y)$-plane[23,24,27]. To maintain the commensurable phase also for the tilted Pythagorean moiré lattice for $\alpha > 0$, as required for the realization of Thouless pumping, the tilt angle $\alpha$, which in our setting plays the role of the velocity of a sliding lattice along the $\mathbf{j}$-direction, must take only discrete values. Namely, it should satisfy $a\alpha k_z/(2\pi) \in \mathbb{N}$, where $a$ is the lattice constant of the square sublattice. Since $V(\boldsymbol{r} - \alpha z\mathbf{j})$ is $a/\alpha$-periodic, for the moiré pattern $I(\boldsymbol{r}, z)$ to be also periodic, with a period $Z$ being multiple of $a/\alpha$, we require that $k_z\alpha^2 Z/(4\pi) \in \mathbb{N}$. For the parameters of our photonic lattice $Z = 2a/\alpha$ and one gets the following expression for the resulting moiré potential

$$U(\boldsymbol{r}, z) = \frac{E_0}{1 + |V(R(\theta)\boldsymbol{r}) + e^{2\pi i(y/a - z/Z)} p V(\boldsymbol{r} - \alpha z\mathbf{j})|^2}. \qquad (2)$$

At any $z > 0$ this potential has transverse periodicity and is determined by the primitive translation vectors $\boldsymbol{e}_1 = a(m\mathbf{i} - n\mathbf{j})$ and $\boldsymbol{e}_2 = a(n\mathbf{i} + m\mathbf{j})$. Thus, the moiré lattice (2) in the $(x, y)$-plane has the lattice constant $L = (m^2 + n^2)^{1/2}a$ and the primitive cell area $(m^2 + n^2)a^2$.

To realize topological transport, the potential $U$ must vary with $z$ adiabatically, a regime that is achieved when $\alpha$ is small enough. Then, the bandgap structure at different values of $z$ determines the light evolution. Fig. 1b illustrates the two upper bands of the tilted moiré lattice (corresponding to $m = 2$ and $n = 1$, i.e., to the first Pythagorean triple $(3, 4, 5)$ used in the experiment) within the first longitudinal period $Z$. The spectrum at a given propagation distance $z$, i.e., the "instantaneous" spectrum, is calculated from the eigenvalue problem $H\varphi_{\nu k}(\boldsymbol{r}, z) = -\beta_{\nu k}(z)\varphi_{\nu k}(\boldsymbol{r}, z)$, where $\varphi_{\nu k}(\boldsymbol{r}, z) = u_{\nu k}(\boldsymbol{r}, z)e^{i\boldsymbol{k}\boldsymbol{r}}$ is the Bloch function corresponding to the band $\nu$ and to the Bloch vector $\boldsymbol{k}$ in the reduced Brillouin zone, $u_{\nu k}(\boldsymbol{r}, z) = u_{\nu k}(\boldsymbol{r} + \boldsymbol{e}, z)$ (here $\boldsymbol{e} = n_1\boldsymbol{e}_1 + n_2\boldsymbol{e}_2$ with $n_{1,2} \in \mathbb{N}$ is an arbitrary lattice vector) is normalized to unity, and $z$ is considered as a parameter. The parameters of the sublattices are chosen to ensure that the first upper gap does not close at any $z$. We also emphasize that it is the relative flatness of the upper band (blue lines in Fig. 1b, c) that leads to the required significant inhibition of diffraction, without the use of any other confining mechanisms. Complete localization, i.e., total suppression of diffraction is possible

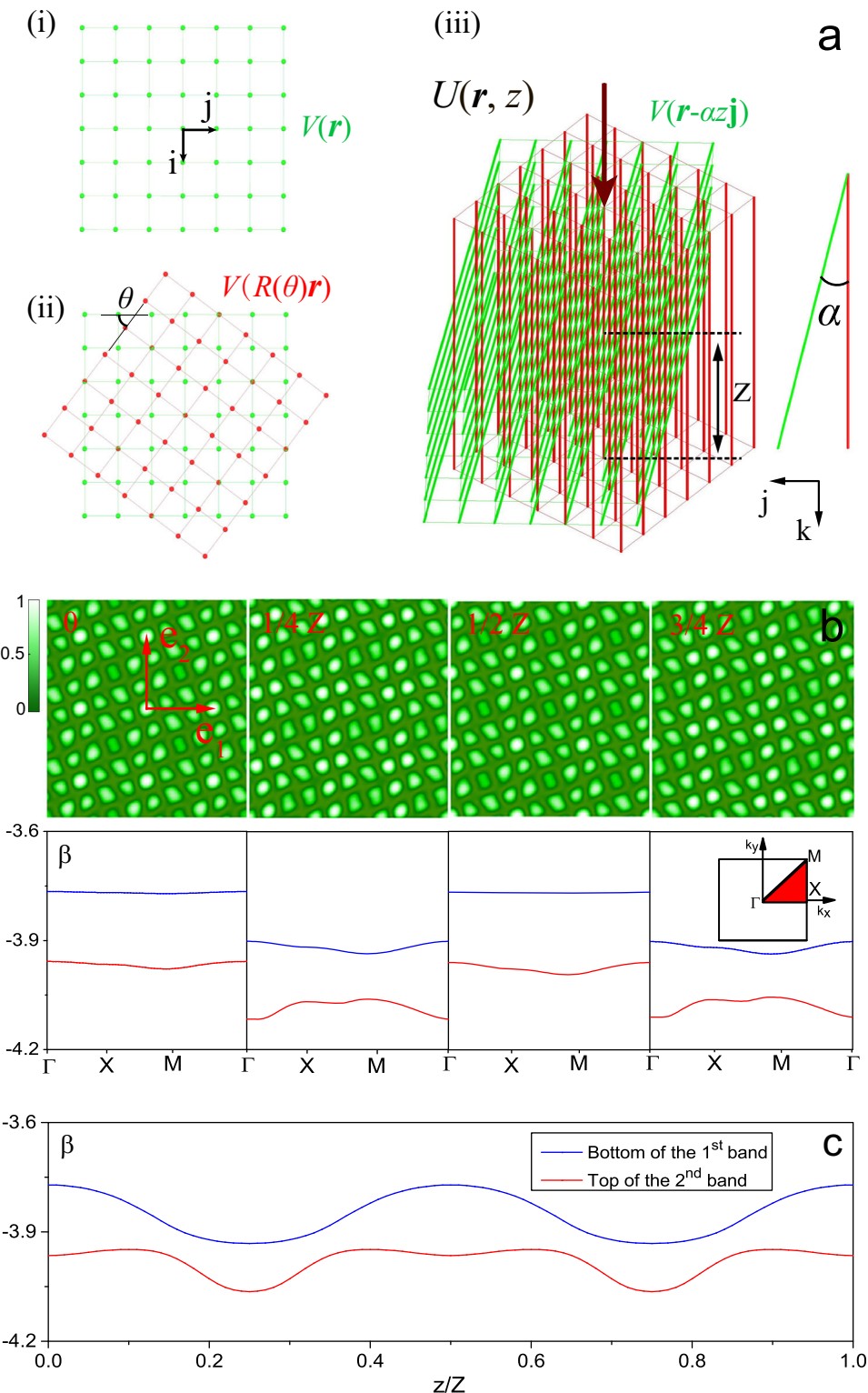

**Fig. 1 | Tilted moiré lattices. a** Schematics of the tilted moiré lattice based on a reference square lattice $V(\boldsymbol{r})$ with primitive translation vectors $a\mathbf{i}$ and $a\mathbf{j}$, where $a$ is the lattice constant (here set to 1) (i). One of the constitutive sublattices is obtained by twisting the reference lattice by the angle $\theta$ around the $z$-axis, yielding the potential $V(R(\theta)\boldsymbol{r})$ (the red-colored lattice in (ii)). The second sublattice is obtained by rotating the reference lattice by an angle $\alpha$ around the $x$-axis (the olive-colored lattice in (iii)), thus producing the potential $V(\boldsymbol{r} - \alpha z\mathbf{j})$. Superimposing $V(\boldsymbol{r} - \alpha z\mathbf{j})$ and $V(R(\theta)\boldsymbol{r})$ results in the moiré potential $U$ shown in (iii). The tilted moiré lattice has a period $Z$ along the direction of the incident beam (shown by a dark-red arrow). **b** Cross-sections of the optically induced moiré lattice $U(\boldsymbol{r}, z)$ with $p = 0.3$ and $E_0 = 7$ (upper row) and the instantaneous bandgap structures (lower row) at distances $z = 0$, $Z/4$, $Z/2$, and $3Z/4$. The red arrows indicate the primitive translation vectors $\boldsymbol{e}_1$ and $\boldsymbol{e}_2$ of the tilted moiré lattice. Blue and red lines in lower row of (**b**) show the first and second bands in the instantaneous lattice spectrum for the same distances. **c** Blue and red lines show the evolution of the bottom edge of the first band and top edge of the second band with distance $z$ on one lattice period $Z$. In (**b**) and (**c**), $\alpha = 0.015$.

only in the incommensurate phases[23,24,27], but partial suppression suffices for our purposes here. In Fig. 1c it is shown that the gap between the first (blue line) and second (red line) bands remains open at all instants of lattice evolution over pumping cycle. These properties ensure that the incident beam initially exciting the upper band will remain in the same band during the subsequent evolution. This result is remarkable, because in contrast to 1D lattices, design of 2D dynamic lattices featuring band structure without band crossings at any instants of evolution period is much more challenging. Thus, for other types of tilted moiré lattices, for example in a lattice corresponding to the Pythagorean triple $(5, 12, 13)$, with all other parameters being equal, band crossing does occur at certain points $z$, as illustrated in Supplementary Fig. 2. The above requirements are not met also in tilted but non-twisted sublattices (propagation is illustrated in Supplementary Fig. 3). This determines the twofold fundamental role of moiré patterns in our experiment: they flatten the bands and introduce non-trivial topology necessary for 2D pumping.

### The two first Chern numbers associated with sliding moiré lattices

Topological Thouless pumping in our setting manifests itself in the transverse displacement of the center of mass (COM) of a localized beam launched at the center of the moiré lattice at $z = 0$. The COM coordinate is defined as $r_c(z) = P^{-1} \int_{\mathbb{R}^2} r |\psi(r, z)|^2 d^2 r$, where $P = \int_{\mathbb{R}^2} |\psi(r, z)|^2 d^2 r$ is the total beam power (it does not change during propagation). To determine the topological characteristics of the pump process and taking into account that the input beam is strongly localized, it is convenient[12,28] to express the COM position in terms of the localized 2D Wannier functions (WFs)[29,30] $w_\nu(r - e, z)$, which are mutually orthogonal and normalized. Namely, the field of the input beam can be expanded as $\psi = \sum_{\nu, e} c_{\nu e}(z) w_\nu(r - e, z)$, where the coefficients $c_{\nu e}(z)$ describe the evolution of the weights of the Wannier functions in the expansion (they are governed by the equations discussed in the Supplementary materials). Because in our experiment the input beam excites mainly the upper band with index $\nu = 1$ we can write: $r_c = \chi(z) R(z) + \rho_t(z) + \rho_{ib}(z)$, where $R(z) = \int_{\mathbb{R}^2} r |w_1(r, z)|^2 d^2 r$ describes the location of the COM in the ideal adiabatic non-diffracting evolution, $\chi(z) = P^{-1} \sum_e |c_{1e}|^2 \lesssim 1$ is the fraction of the input beam power contained in the upper band (in our experiment $\chi(z)$ remains close to one; in practice it can never be made one exactly), while $\rho_t(z)$ and $\rho_{ib}(z)$ describe respectively the integral deviation of the COM trajectory due to the effect of inter-site tunneling in the upper band and inter-band transitions, resulting in the energy redistribution among different Wannier states. Even if the input beam excites a single Wannier state and the inter-band transitions are suppressed (i.e., $|\rho_{ib}|$ is negligible), the diffraction mediated by inter-site tunneling results in the redistribution of the power among Wannier states of the upper band, which are localized at different lattice cites. This results in the characteristic pattern of the output signals (see Figs. 2e–g and 3d, below). Importantly, the $R(z)$ term in this expression is of topological nature, in the sense that its variation over one $z$-period is quantized and is given by $R(Z) = C_1 e_1 + C_2 e_2$, where the first Chern numbers $C_j, j = 1, 2$, are defined by (see Supplementary materials)

$$C_j = i \int_0^Z dz \int_{-\pi/L}^{\pi/L} \frac{dk_j}{2\pi} \left( \left\langle \frac{\partial u_{1k}}{\partial z} \Big| \frac{\partial u_{1k}}{\partial k_j} \right\rangle - \left\langle \frac{\partial u_{1k}}{\partial k_j} \Big| \frac{\partial u_{1k}}{\partial z} \right\rangle \right). \quad (3)$$

The angular brackets denote integration over the area of the transverse primitive cell of the tilted moiré lattice. For the potentials used in the experiment with $m = 2$ and $n = 1$, the Chern numbers take the values $C_1 = -1$ and $C_2 = 1$. Note that since our moiré lattice is non-separable, other choice of twisting angle may give other values of $C_1$ and $C_2$. In particular, changing twisting angle from $\theta$ into $-\theta$ results in change of the sign of $C_1$. If instead of the lattice corresponding to $(m, n) = (2, 1)$ we take the lattice generated for opposite twisting angle and

corresponding to $(m, n) = (-2, 1)$, we can observe that $(C_1, C_2)$ changes from $(-1, 1)$ into $(1, 1)$, and this, of course, results in modification of the pumping direction of light, see Supplementary Fig. 5. We emphasize that, while apparently simple final expression for the quantized displacement $R(Z)$ appears to be decomposed into two orthogonal components, the pumping is intrinsically 2D, i.e., shifts along $x$ and $y$ axes are mutually interrelated because the optical potential (2) cannot be represented as a sum of two simple 1D potentials, and the two Chern number, $C_1$ and $C_2$, are not independent.

### Topological Thouless pumping of 2D light beams

These predictions are compared in Fig. 2 with the outcomes of the direct numerical evolution governed by Eq. (1) with an input Gaussian beam $\psi(r) = \exp(-r^2/r_0^2)$ of width $r_0 = 0.7$ that covers approximately one cell of the lattice. We first consider a very small tilt angle $\alpha = 0.002$, well within the adiabatic variation of the moiré lattice. In Fig. 2a we show the evolution of the numerically calculated displacement of the beam center position $|r_c(z)|$ over two $z$-periods. The trajectory of the COM of the beam in the transverse plane is depicted in Fig. 2b. The dashed lines in Fig. 2a, b show the predicted average trajectory described by the vector $R_{av}(z) = (z/Z)(C_1 e_1 + C_2 e_2)$ that is determined by the two first Chern numbers. It is clearly visible that the trajectory is different from the direction of tilting of one sublattice with respect to another one, shown in panels (b) and (d) by the dotted line with an arrow. Thus, the crucial difference of our results and those reported in previous experiments[10] is readily apparent: the direction of the beam COM displacement is not limited to small angles with respect to the direction of pumping. Instead, it is characterized by a large angle (18° in this particular case), and can be varied in a discrete manner by changing the order of the Pythagorean triple and by adjusting properly all other parameters. We also note that one observes nearly perfect agreement between the numerical results (solid lines) and the ideal expectations (dashed lines), both for the direction and magnitude of the COM displacement, in spite of the beam diffraction, visible in Fig. 2e. Somewhat irregular deviations of the COM trajectory from its average value in Fig. 2a, b are due to the evolution of $\rho_t(z) + \rho_{ib}(z)$ that is caused by inter-site and inter-band power exchange.

The above results correspond to the nearly adiabatic regime that occurs for sufficiently small values of the tilt angle $\alpha$. Although such angles still do not correspond to the mathematical adiabatic limit, we nevertheless observed an excellent agreement between the direction/magnitude of the beam pumping and prediction based on the topology of the system. A comparison of the pumping of the beam in moiré lattices for progressively decreasing angles $\alpha$ (illustrating gradual transition to the adiabatic regime) is given in Supplementary Fig. 6. At the same time, the smaller the value of $\alpha$ the longer the sample needed for the observation of the effect. To elucidate experimental conditions achievable with currently available photo-refractive crystal dimensions, one has to consider larger tilt angles. The comparison of the numerical results with predictions for pumping at $\alpha = 0.015$ (this angle is used in our experiments, enabling almost 1.7 pumping cycles on the 2 cm length of our SBN sample) is shown in Fig. 2c, d, f, g. One still observes an accurate prediction for the direction of the COM displacement (Fig. 2d), while the magnitude of the displacement $|r_c(z)|$ is now somewhat smaller than it would be for an ideal pumping $|R_{av}(z)|$ (Fig. 2c). The deviation is explained by the diffraction and excitation of lower bands by the input Gaussian beam (corresponding to $\chi(z) < 1$). If disregarding radiation below some intensity level (we set it to be 9% of the peak intensity) or considering the trajectory of the peak intensity we recover the theoretical formula for $|R_{av}(z)|$ with good precision even in the case $\alpha = 0.015$ (see Supplementary Fig. 4). Because the first band remains well separated from the second band at all distances (Fig. 1c), the scaling factor $\chi(z)$ only weakly changes with $z$ and on average one observes the displacement $|r_c(z)| \propto z$ (Fig. 2c).

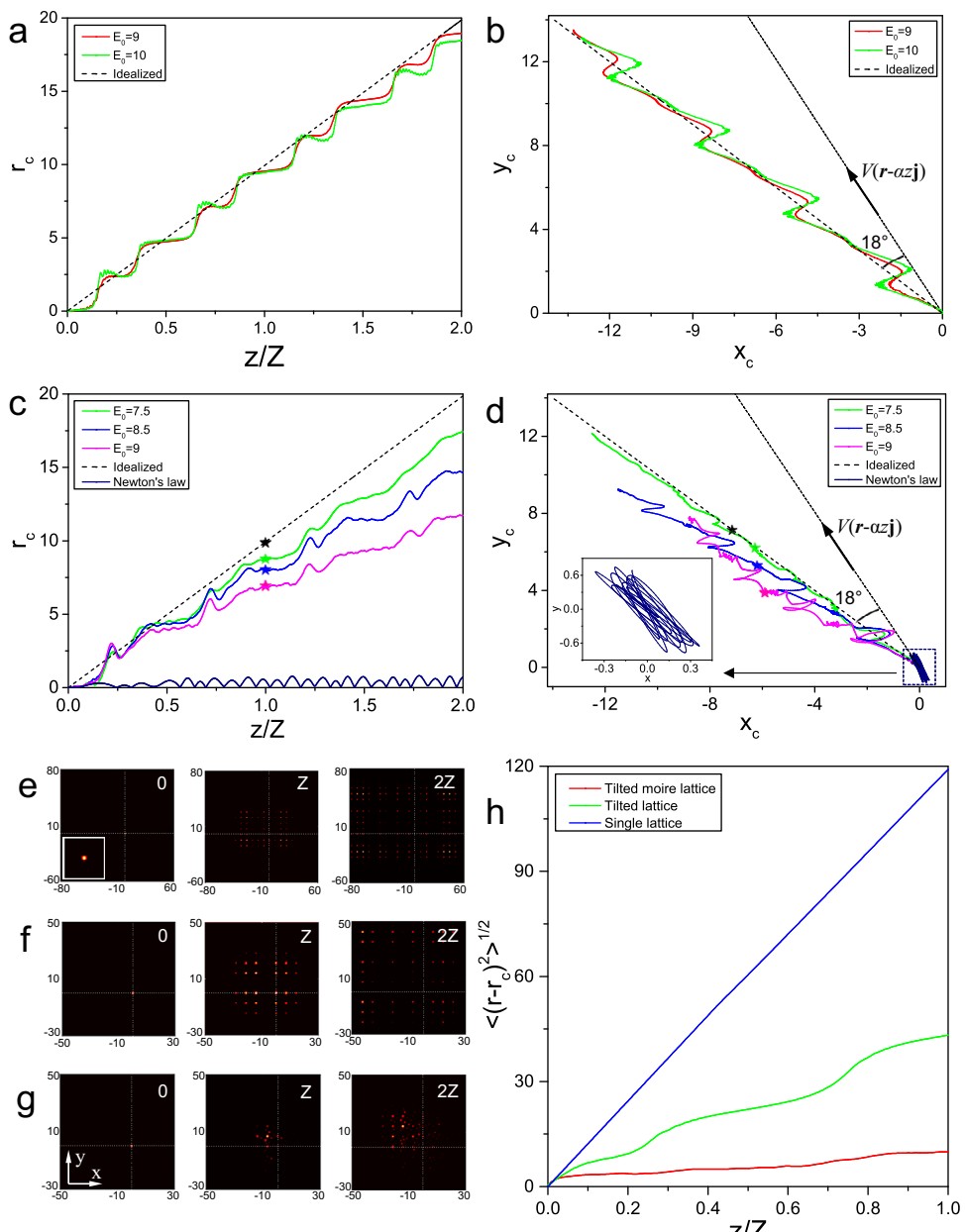

**Fig. 2 | Numerical simulation of Thouless pumping in tilted moiré lattices.** Pythagorean angle $\theta = \arctan(4/3)$ ($m = 2$, $n = 1$) and tilt angles $\alpha = 0.002$ (**a**, **b**, **e**) and $\alpha = 0.015$ (**c**, **d**, **f**, **g**). **a**, **c** Displacements of the light beam COM, $|r_c|$, with distance $z$. **b**, **d** Trajectories of the COM $r_c = (x_c, y_c)$ in the transverse plane. The asterisks correspond to the points $z = Z$, i.e., to one longitudinal lattice period. Lines of different color correspond to different amplitudes $E_0$ of the moiré lattice. Dashed diagonal lines show the evolution of $R_{av}(z)$ in (**b**) and (**d**), and of its modulus in (**a**) and (**c**) upon propagation. The results for classical motion in the potential $U$ are

shown by the dark-blue solid lines. The arrows along dotted lines in (**b**) and (**d**) show the direction of tilting of one sublattice with respect to another one. **e** Snapshots of beam propagation for $\alpha = 0.002$ and $E_0 = 10.0$ (zoom of the initial intensity distribution is shown within white square). **f**, **g** Snapshots of beam propagation for $\alpha = 0.015$ and $E_0 = 7.5$ and $E_0 = 9.0$, respectively. **h** Evolution of the mean radius of the beam propagating inside tilted moiré lattice (red line), a tilted lattice obtained without twisting of two sublattices (green line) with $p = 0.3$, and in a single reference lattice (blue line) with depth $p = 1.3$ for $\alpha = 0.015$. In all cases, $a = \pi$, $p = 0.3$.

To expose the nature of the transport, we use the Ehrenfest theorem and compare the motion of the beam COM with a trajectory of the corresponding virtual classical particle governed by Newton's second law, $d^2r/dz^2 = \nabla U(r, z)$, with the initial conditions $r(0) = \mathbf{0}$ and $(dr/dz)_{z=0} = \mathbf{0}$. The selected relative depth $p$ of the second sublattice is such that the central local maximum of the optical potential (2) persists and remains close to its initial location upon transformation of the lattice with $z$ (Fig. 1b). This generates the classical trajectory depicted in Fig. 2c, d. It is readily visible that such a fictitious particle undergoes oscillations around the local maximum of the potential but does not exhibit directional transport.

To further confirm the topological nature of the pumping, we check the influence on it of the lattice depth $E_0$ (applied dc field). In the adiabatic regime with $\alpha = 0.002$ (Fig. 2a, b) the COM displacement is essentially independent of $E_0$. At $\alpha = 0.015$ the effect of the potential depth, shown in Fig. 2c, d, is more pronounced that is attributed to non-adiabatic effects leading to the dependence of $\chi(z)$ on $E_0$ (for the role of the non-adiabatic effects in Thouless pumping see ref. 31). In all analyzed cases we observe discrete diffraction patterns (Fig. 2e–g). The spatial extent of these patterns decrease with $E_0$. It may be larger than or comparable with the displacement of the COM for a shallower (Fig. 2f) and deeper (Fig. 2g) lattices, respectively. However, in spite of being clearly detectable due to very long propagation distances

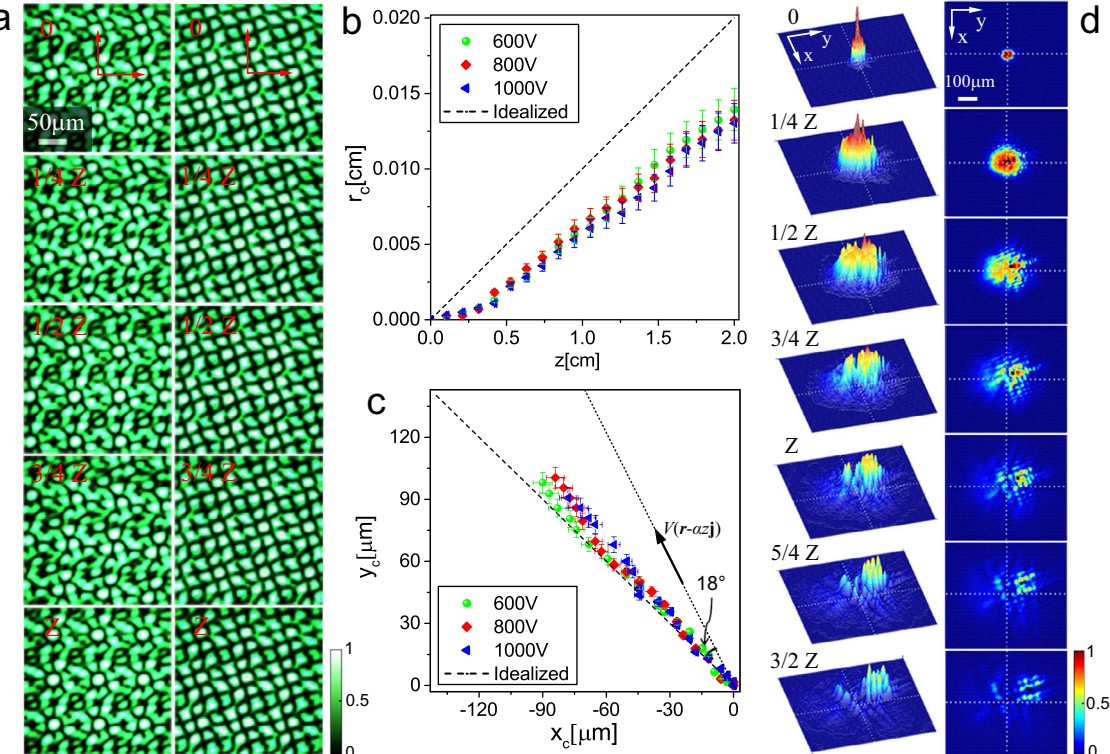

**Fig. 3 | Experimental observation of two-dimensional Thouless pumping.**
**a** Optically induced moiré lattices with $I_2/I_1 = 1.0$ (left column) and $I_2/I_1 = 0.1$ (right column) at different propagation distances inside the sample. Panels **b–d** correspond to the right column. **b** Measured COM displacement $|\mathbf{r}_c|$ with $z$. **c** The measured trajectory $\mathbf{r}_c = (x_c, y_c)$ in the transverse plane for various voltages applied to the crystal. The arrow along dotted lines shows the direction of the sliding of one sublattice with respect to another one. **d** Surface and contour plots showing signal beam intensity distributions at different distances inside the sample at $U_{bias} = 600$ V. The $x$ and $y$ axes in these distributions are determined by the directions of the primitive translation vectors $\mathbf{e}_1$ and $\mathbf{e}_2$.

corresponding to a full cycle of the adiabatic evolution, one concludes that the diffraction is indeed strongly suppressed due to the flatness of the bands of the spectrum of the moiré lattices. We emphasize this by comparing in Fig. 2h the root mean square widths, $\langle(\mathbf{r} - \mathbf{r}_c)^2\rangle^{1/2} = [P^{-1}\int_{\mathbb{R}^2}(\mathbf{r} - \mathbf{r}_c)^2|\psi|^2 d^2 r]^{1/2}$, of the beam propagating in the moiré lattice used in experiments (red line), in the tilted lattice without twist, i.e., at $\theta = 0$ (green line), and in the non-tilted lattice with $\alpha = \theta = 0$ (blue line). We observe the reduction of diffraction in moiré lattice by an order of magnitude.

**Experimental observation of 2D topological Thouless pumping of light beams**

As indicated above, we realized experimentally the Thouless pumping with optically induced lattices in photorefractive SBN:61 crystals. Each of two sublattice beams, whose interference produces the moiré potential (2) in the volume of the photorefractive crystal, was created by four properly tilted ordinary-polarized plane waves (see Methods and Supplementary Fig. 1). The relative amplitude $p$ of the second sublattice was controlled by tuning the intensity ratio of the sublattice-forming beams $I_2/I_1$, namely, $p = \sqrt{I_2/I_1}$. The intensity of the first sublattice $I_1 = 7$ mW/cm² was fixed in the experiment, while varying the ratio of beam intensities to $I_2/I_1 = 0.1$ allowed to reproduce approximately the case with $p = 0.3$ considered above in the numerical simulations. Examples of the created tilted moiré lattices with $\theta = \arctan(4/3)$ and $\alpha = 0.015$ are presented in Fig. 3a. Lattices are depicted for several propagation distances within one longitudinal period $Z$. To further illustrate that patterns are replicated after each longitudinal period, we present them not only for $I_2/I_1 = 0.1$ (right column), but also for the

case of $I_2/I_1 = 1.0$ (left column) with more pronounced local maxima in the lattice profile. To observe the 2D Thouless pumping, an extraordinarily polarized Gaussian beam from a continuous wave He-Ne laser ($\lambda = 632.8$ nm) that serves as a low-power signal, was launched normally into the front facet of the crystal. The diameter of the Gaussian beam was about 18 μm, covering approximately one channel site (i.e., a local index maxima) of the lattice. The propagation of the signal beam through the sample was monitored and the location of its COM was measured. Since the crystal has a transverse dimensions of 5 mm × 5 mm and the signal beam may expand considerably in the course of discrete diffraction, we applied to the sample a biasing voltage exceeding $U_{bias} = 600$ V, that approximately corresponds to $E_0 = 9.0$ in simulations (Fig. 2g).

The experimentally measured displacement of the COM of the signal beam as well as its trajectory in the transverse plane are presented in Fig. 3b, c, along with dashed lines corresponding to the theoretical prediction for $\mathbf{R}_{av}(z)$. The COM displacement substantially exceeds the input beam width, while its direction perfectly agrees with theory. Its magnitude is slightly lower than the value expected for the fully adiabatic conditions because of the reasons explained above, and anyway it is in perfect agreement with the results of the numerical modeling presented in Fig. 2d. Practical independence of the COM displacement and trajectory of the applied voltage is consistent with the topological nature of the transport. The measured intensity distributions at different propagation distances in Fig. 3d (the direction of the $x$ and $y$ axes in this figure are determined by the primitive translation vectors $\mathbf{e}_1$ and $\mathbf{e}_2$) are in excellent agreement with the numerically calculated intensity patterns shown in Fig. 2g. The data clearly illustrate the considerable COM displacement, comparable with the width of the discrete diffraction pattern.

## Discussion

We obtained experimental evidence of fully 2D topological Thouless pumping of light using a new type of purpose-designed tilted photonic moiré lattice. The lattices are optically induced structures that remain periodic upon propagation in the transverse direction and at the same time periodically reproduce their shapes in the longitudinal direction. We showed that in spite of diffraction that was greatly suppressed by virtue of the flat bands afforded by the moiré lattice, and also in spite of unavoidable in an experiment non-adiabatic effects, the motion of the center of mass of a paraxial beam in such lattices is determined by the global topology of the lattice bands, rather than by the local details of the lattice shape. Crucially, it is well described by two topological invariants—the first Chern numbers. We stress that moiré lattices are induced by interfering plane waves in the entire illuminated volume of the photorefractive crystal in one simple step, which is clearly beneficial in comparison with pumping schemes based on waveguide arrays, where delicate control of the separations between waveguides along the entire sample length is required[5–7]. Also, unlike such previous pumping schemes relying on transformations between localized edge and bulk states, the pumping scheme here is implemented with 2D localized beams propagating in the bulk of the lattice. Furthermore, the moiré patterns used here are highly flexible and provide a powerful tool to study topological transport in diverse geometries, which are not limited to the tetragonal Bravais lattices addressed here and that include also the regime where the nonlinear and the nonlocal character of the photorefractive response becomes noticeable. Furthermore, the exquisite control afforded by the employed setting opens the possibility to explore topological transport associated with higher-order bands, as well as transport in incommensurate lattices, therefore making accessible the so far unexplored possibility of topological transport at the transition between periodicity and aperiodicity of the underlying system. The setting reported in this work can be straightforwardly applied in atomic and acoustic systems, beyond optical applications.

## Methods

The experimental setup is illustrated in Supplementary Fig. 1. The lattice was created using optical induction technique, as described in ref. 25 and first realized experimentally in ref. 26. A cw frequency-doubled Nd:YAG laser at wavelength $\lambda = 532$ nm is used to "write" lattices in a biased photorefractive crystal (SBN: 61). The crystal has dimensions $5 \times 5 \times 20$ mm³, and the 20 mm direction defines the optical axis (see the dashed line in the beam path **1**). To generate a moiré lattice, an amplitude mask 1 with two groups of $2 \times 2$ pinholes is used, as shown in the figure inset, where the first group is denoted in green and the second group in yellow. The second group is twisted by an angle $\theta$ with respect to the first group, and, as a result, the two square-lattice-forming beams produced by each group, when combined together, form a moiré lattice with a twisting angle $\theta$. However, in order to create the tilted moiré lattice, pinholes of the second group were additionally displaced as a whole from the pinholes of the first group by a distance $d$, thus, after being refracted by the L2, the second-lattice-forming beams make a tilt angle $\alpha$ with respect to the first-lattice-forming ones, $\alpha = d/f_2$, where $f_2$ is the focal length of L2. According to the quantization condition for angle $\alpha = 2\pi m/ak_z$ (see the main text), the displacement $d$ must be quantized too, so that $d = 2\pi m f_2/ak_z = m k_z f_2/k_z$. By noting that $k_x/k_z = d_0/f_2$, with $d_0$ being the distance of the pinhole from the optical axis, one finds $d = m d_0$. $m = 1$ is used throughout this work. The experimental error for determining $\alpha$ is 0.75 mrad, combing the measurement error of distance $d$ and the focal length $f_2$. For our working angle $\alpha = 0.015$, this error does not exceed 5% from exact $\alpha$ value.

In order to fine-tune the relative amplitude strength between two lattice-forming beams, a second mask (Mask 2) made of a polarizer film, superimposed with a half-wave plate and a polarizer is used. The two lattice-creating beams after being tuned in this way are further modulated by a half-wave plate before entering the SBN crystal, to ensure that they are ordinarily polarized.

Beam path **2** corresponds to an extraordinary-polarized probe beam with wavelength $\lambda = 632.8$ nm, used for studying light propagation through the induced lattice. An imaging lens and a CCD camera are used to record the light intensity patterns, for both the lattice-forming beams and signal light.

## Data availability

The data that support the findings of this study are available from the corresponding author upon reasonable request.

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

## Acknowledgements

P.W., Q.F., R.P. and F.Y. acknowledge support from the NSFC (No. 91950120), Scientific funding of Shanghai (No.9ZR1424400), and Shanghai Outstanding Academic Leaders Plan (No.20XD1402000). Y.V.K. and L.T. acknowledge support from the Severo Ochoa Excellence Programme (CEX2019-000910-S), Fundacio Privada Cellex, Fundacio Privada Mir-Puig, and CERCA/Generalitat de Catalunya. V.V.K. acknowledges financial support from the Portuguese Foundation for Science and Technology (FCT) under Contracts PTDC/FIS-OUT/3882/2020 and UIDB/00618/2020.

## Author contributions

P.W. and Q.F. contributed equally to this work. All authors (P.W., Q.F., R.P., Y.V.K., L.T., V.V.K. and F.Y.) contributed significantly to the paper.

## Competing interests

The authors declare no competing interests.
