## [Peer review file · Nature Communications]

REVIEWER COMMENTS

Reviewer #1 (Remarks to the Author):

In their work „Two-dimensional Thouless pumping of light in photonic moire lattices,“ the authors explore exactly this in a novel structure that was induced in photorefractive crystals.

I like this work a lot: the idea is nice, the implementation clever, and the results convincing. As topological photonics is a hot topic such that this manuscript will potentially experience much interest of a wider audience, I in principle recommend publication in Nature Communications.

However, before acceptance I would like to raise three minor questions; I'd appreciate if the authors could respond to that.

1) It is well known that the realization of ideal Thouless pumping requires an adiabatic variation of the potential. What are the experimental limitations in this respect? Can the authors comment on the required experimental lengths and tilt angles that would allow approaching the adiabatic regime even closer?

2) I would appreciate some explicit equations for the coefficients c_{1e} , which depend on z , from the expansion of the field in the basis of Wannier functions. If there is no room for this in the main text, these equations could be added into Supplementary Material.

3) When inspecting Fig. 1, one can see that the bandgap between bands 1 and 2 is not so large. Could it happen that due to this light can tunnel to band 2 during propagation?

After these questions are addressed I am in full support of acceptance of this nice work.

Reviewer #2 (Remarks to the Author):

The paper reports a novel experimental realization of driving an optical beam by means of cyclic modulations of the lattice-induced bandgap spectrum (alias topological pumping of the beam). The results are interesting because they offer a new experimental emulation of an important quantum effect, realized by means of classical linear optics. The experimental findings are well predicted by simulations of the corresponding paraxial-propagation equation, which includes the effective photonic potential.

The results are appropriate for the publication, but it is recommended to clarify a few essential points. First, it is stressed in the paper that the Thouless-pumping mechanism is realized in the framework of the adiabatic regime of the cyclic modulations. However, an actual criterion of the adiabaticity is not given in an explicit form. As concerns effects of violation of the adiabaticity, it is mentioned that they were addressed in Ref. [31], but it is not clearly explained if such effects are observed in the present work, and how they may manifest themselves. Furthermore, while the text repeatedly stresses the importance of the adiabaticity and flatness of the spectral bands, it is unclear if these two principles are mutually linked, or they are completely independent. Incidentally, how many actual cycles of the pumping are actually realized in the theoretical and experimental parts of the work?

Another question pertains to the assumption that the ratio of propagation scales, which is written, sans number, above Eq. (2), must be an integer. What may be an acceptable range of deviations from this condition, which would still allow one to observe the topological-pumping regime? And how accurate does this condition hold in the experiment?

Lastly, it is recommended to improve the abstract, which does not explicitly mention the systematic simulations, referring solely to the experimental realization of the effect. Then, the reader may be surprised by the fact that introduction is followed by detailed simulations, which are not mentioned in the abstract. Of course, the accurate simulations are an essential asset of the paper, but this asset should not appear as a surprise.

Replies to Referee #1

1. **Comment:** In their work, “Two-dimensional Thouless pumping of light in photonic moiré lattices,” the authors explore exactly this in a novel structure that was induced in photorefractive crystals. I like this work a lot: the idea is nice, the implementation clever, and the results convincing. As topological photonics is a hot topic such that this manuscript will potentially experience much interest of a wider audience, I in principle recommend publication in Nature Communications.

Reply: We thank this Referee for positive recommendation and for his useful comments that allowed us to improve the presentation of our results.

2. **Comment:** It is well known that the realization of ideal Thouless pumping requires an adiabatic variation of the potential. What are the experimental limitations in this respect? Can the authors comment on the required experimental lengths and tilt angles that would allow approaching the adiabatic regime even closer?

Reply: To observe perfect pumping over one cycle one must control the filling of the bands and the pumping process has to be adiabatic, i.e., ideally the pumping cycle (z-period of the lattice) should tend to infinity. However, in reality the sample, where moiré lattice is induced, has a finite length (2 cm long in our case), which puts a practical limitation with respect to how close one can approach the adiabaticity in typical experiments on photorefractive crystal platform. Please note, that such practical limitations exist in any experimental verification of the adiabatic transport. In our experiment, we used a tilt angle $\alpha=0.015$, which leads to 1.7 pumping cycles (1.7 z-periods of the lattice) on 2 cm length of our sample. Even though this is still not the mathematical adiabatic limit, nevertheless we observed a remarkable agreement between the direction of beam pumping and prediction based on the topology of the system. In other words, approaching even closer to the adiabatic limit, will not change the observed effect. To prove this, we added into Supplementary Material the results of simulations of the displacement of the beam center of mass upon pumping in moiré lattice for several progressively decreasing angles α (results for two such angles are included in the main text in Fig. 2) at fixed dc field E_0 . These results clearly show that the pumping regime becomes nearly adiabatic (i.e., even COM displacement for the light beam practically exactly coincides with prediction based on the Chern numbers) in samples with lengths ~ 5 -10 cm that in principle can be manufactured. Corresponding discussion is added in the main text, at the bottom of page 8 and top of page 9, as well as in new section in Supplementary Material.

3. **Comment:** I would appreciate some explicit equations for the coefficients $c_{\{1e\}}$, which depend on z, from the expansion of the field in the basis of Wannier functions. If there is no room for this in the main text, these equations could be added into Supplementary Material.

Reply: We have followed this advice of the Referee and included the equations for coefficients c_{1e} into Supplementary Material in the paragraph “Tight-binding approximation”.

4. **Comment:** When inspecting Fig. 1, one can see that the bandgap between bands 1 and 2 is not so large. Could it happen that due to this light can tunnel to band 2 during propagation?

Reply: Indeed, this is a crucial issue: if appreciable tunneling between bands would occur, then the light propagation dynamics would be determined not only by the Chern number of the initially excited first band, but also by the Chern number of the second band. This is expected to be the case of the non-Abelian Thouless pumping. In our case pumping regime is not too far from adiabatic one (please see the reply to the comments above) and on this reason no significant tunneling between bands occurs in the course of evolution (simulations reported in Figure 2 for $\alpha=0.002$ shows that the maximal population of the second band is weighted to be less than 5%, in comparison with 95% population of the first, excited band). This is also confirmed by the excellent agreement between theory and the experimental/numerical results on the pumping distance/direction of the light beam.

Replies to Referee #2

1. **Comment:** The paper reports a novel experimental realization of driving an optical beam by means of cyclic modulations of the lattice-induced bandgap spectrum (alias topological pumping of the beam). The results are interesting because they offer a new experimental emulation of an important quantum effect, realized by means of classical linear optics. The experimental findings are well predicted by simulations of the corresponding paraxial-propagation equation, which includes the effective photonic potential. The results are appropriate for the publication, but it is recommended to clarify a few essential points.

Reply: We thank the Referee for carefully assessing our work and for his useful comments that allowed us to clearly improve the presentation of our results.

2. **Comment:** First, it is stressed in the paper that the Thouless-pumping mechanism is realized in the framework of the adiabatic regime of the cyclic modulations. However, an actual criterion of the adiabaticity is not given in an explicit form. As concerns effects of violation of the adiabaticity, it is mentioned that they were addressed in Ref. [31], but it is not clearly explained if such effects are observed in the present work, and how they may manifest themselves.

Reply: A perfect pumping could be achieved when the pumping cycle (z-period of the lattice $Z=2a/\alpha$) tends to infinity. In reality, the sample length is finite (2 cm long in our case) that imposes a practical limitation with respect to how close one can approach the adiabaticity. In our experiment we used an angle $\alpha=0.015$, yielding lattice period of 1.2 cm, which is of course not very close to the mathematical adiabatic limit, but still, we observed an excellent agreement between the direction of the beam pumping and prediction based on

the topology of the system. This confirms that we are working in the quasi-adiabatic regime. To illustrate how pumping dynamics depends on the angle α (z-period of the lattice) and to give an idea on the transition to the adiabatic regime we added into Supplementary Material the results of simulations of the displacement of the beam center of mass upon pumping in moiré lattice for several progressively decreasing angles α (results for two such angles are included into the main text in Fig. 2) at fixed dc field E_0 with the indication of corresponding z-periods of the lattice. These results clearly show that pumping regime becomes nearly adiabatic (i.e. even COM displacement for light beam practically exactly coincides with prediction based on Chern number) in samples with lengths ~ 5 -10 cm that in principle can be manufactured. On the other hand, when cyclic modulation becomes too fast, it would result in significant inter-band transitions. In this regime the direction/magnitude of displacement of the beam depends on the details of modulation (rather than determined by topological indices of the bands), as explained in Ref. [31]. We have added discussion of this point at the bottom of page 8 and at the top of page 9, as well as in new section in Supplementary Materials.

3. **Comment:** Furthermore, while the text repeatedly stresses the importance of the adiabaticity and flatness of the spectral bands, it is unclear if these two principles are mutually linked, or they are completely independent. Incidentally, how many actual cycles of the pumping are actually realized in the theoretical and experimental parts of the work?

Reply: These two ingredients of our study – adiabaticity and flatness of the bands – are independent ingredients. Adiabaticity guarantees the absence of significant inter-band tunneling in the course of propagation, so that pumping process can be described only by the Chern numbers for the first excited band, as specified above (please also see the response to the first Referee). Band flatness – representative for moiré lattices – in turn, allows us to substantially suppress diffraction broadening of the propagating beams (that otherwise would be very considerable and would hide the effect, see the example of propagation dynamics for two untwisted/untilted sublattices presented in Supplementary Material). Thus, since our purpose is to demonstrate clear pumping of a two-dimensional light beam in a bulk system without relying on the confinement due to artificial edges or nonlinear effects, we achieved suppression of diffraction by creating z-periodic moiré structure. We clarified this point on page 4, line 5-14. For the tilt angle of $\alpha=0.015$ used in our experiments, the pumping period is $Z=1.2$ cm, i.e. there are ~ 1.7 pumping cycles on the 2 cm length of our SBN sample. This is now clearly stated on page 9, line 5-6. Note also we have stressed the importance of these aspects on page 4 of the manuscript.

4. **Comment:** Another question pertains to the assumption that the ratio of propagation scales, which is written, sans number, above Eq. (2), must be an integer. What may be an acceptable range of deviations from this condition, which would still allow one to observe the topological-pumping regime? And how accurate does this condition hold in the experiment?

Reply: We thank the Referee for this highly pertinent and nontrivial question. Above Eq. (2) we formulate two conditions on α . The first one ensures the exact periodicity along the

z-axis. The second one ensures the periodicity along the y-axis. To address this question of the Referee in the Supplementary Material we show Fig. 6, where small α deviating from the exactly quantized values are considered numerically. One can appreciate small deviations from the exact quantization conditions for α . Please notice that we are talking about very long distances, with respect to which the observed deviations are indeed very small.

In Methods Section and also in Supplementary Figure 1, we described how we experimentally generate lattices satisfying the exact quantization conditions for angle α . The error may come from the fabrication of the Masks 1 and 2, and the measurement of the focal length of Lens 2 and distance d . This can give an error in definition of angle α not exceeding ± 0.75 mrad (5 % from the working angle $\alpha=0.015$). We added a statement about this in the Methods section.

5. **Comment:** Lastly, it is recommended to improve the abstract, which does not explicitly mention the systematic simulations, referring solely to the experimental realization of the effect. Then, the reader may be surprised by the fact that introduction is followed by detailed simulations, which are not mentioned in the abstract. Of course, the accurate simulations are an essential asset of the paper, but this asset should not appear as a surprise.

Reply: We modified the abstract of the manuscript, where we now mention that our experimental results are fully supported by systematic numerical simulations in the frames of the Schrödinger equation governing propagation of the probe beam in optically-induced photorefractive moiré lattice.

REVIEWERS' COMMENTS

Reviewer #1 (Remarks to the Author):

The authors implemented all suggestions of both referees. I am fully satisfied. I congratulate the authors for this very nice piece of work and support publication of their results in Nature Communications.

Reviewer #2 (Remarks to the Author):

The revised text has adequately implemented comments from the original review. The resubmitted paper may be recommended for the publication.

“Two-dimensional Thouless pumping of light in photonic moiré lattices” (manuscript ID NCOMMS-22-21837A)

Replies to Referee #1

Comment: The authors implemented all suggestions of both referees. I am fully satisfied. I congratulate the authors for this very nice piece of work and support publication of their results in Nature Communications.

Reply: We thank this Referee for positive recommendation and for his useful comments raised in the first round that allowed us to improve the presentation of our results.

Replies to Referee #2

Comment: The revised text has adequately implemented comments from the original review. The resubmitted paper may be recommended for the publication.

Reply: We thank this Referee for positive recommendation and for his useful comments raised in the first round that allowed us to improve the presentation of our results.